# DPSNet: End-to-end Deep Plane Sweep Stereo

**Sunghoon Im**[*1]**, Hae-Gon Jeon**[2]**, Stephen Lin**[3]**, In So Kweon**[1]

[1] KAIST, [2] Carnegie Mellon University, [3] Microsoft Research Asia

dlarl8927@kaist.ac.kr, haegonj@andrew.cmu.edu,
stevelin@microsoft.com, iskweon77@kaist.ac.kr

## Abstract

Multiview stereo aims to reconstruct scene depth from images acquired by a camera under arbitrary motion. Recent methods address this problem through deep learning, which can utilize semantic cues to deal with challenges such as textureless and reflective regions. In this paper, we present a convolutional neural network called DPSNet (Deep Plane Sweep Network) whose design is inspired by best practices of traditional geometry-based approaches for dense depth reconstruction. Rather than directly estimating depth and/or optical flow correspondence from image pairs as done in many previous deep learning methods, DPSNet takes a plane sweep approach that involves building a cost volume from deep features using the plane sweep algorithm, regularizing the cost volume via a context-aware cost aggregation, and regressing the dense depth map from the cost volume. The cost volume is constructed using a differentiable warping process that allows for end-to-end training of the network. Through the effective incorporation of conventional multiview stereo concepts within a deep learning framework, DPSNet achieves state-of-the-art reconstruction results on a variety of challenging datasets.

## 1 Introduction

Various image understanding tasks, such as semantic segmentation Couprie et al. (2013) and human pose/action recognition Shotton et al. (2011); Wang et al. (2016), have been shown to benefit from 3D scene information. A common approach to reconstructing 3D geometry is by multiview stereo, which infers depth based on point correspondences among a set of unstructured images Hartley & Zisserman (2003); Schönberger et al. (2016). To solve for these correspondences, conventional techniques employ photometric consistency constraints on local image patches. Such photo-consistency constraints, though effective in many instances, can be unreliable in scenes containing textureless and reflective regions.

Recently, convolutional neural networks (CNNs) have demonstrated some capacity to address this issue by leveraging semantic information inferred from the scene. The most promising of these methods employ a traditional stereo matching pipeline, which involves computation of matching cost volumes, cost aggregation, and disparity estimation Flynn et al. (2016); Kendall et al. (2017); Huang et al. (2018); Chang & Chen (2018). Some are designed for binocular stereo Ummenhofer et al. (2017); Kendall et al. (2017); Chang & Chen (2018) and cannot readily be extended to multiple views. The CNN-based techniques for multiview processing Flynn et al. (2016); Huang et al. (2018) both follow the plane-sweep approach, but require plane-sweep volumes as input to their networks. As a result, they are not end-to-end systems that can be trained from input images to disparity maps.

In this paper, we present Deep Plane Sweep Network (DPSNet), an end-to-end CNN framework for robust multiview stereo. In contrast to previous methods that employ the plane-sweep approach Huang et al. (2018); Flynn et al. (2016), DPSNet fully models the plane-sweep process, including construction of plane-sweep cost volumes, within the network. This is made possible through the use of a differentiable warping module inspired by spatial transformer networks Jaderberg et al. (2015) to build the cost volumes. With the proposed network, plane-sweep stereo can be learned in an end-to-end fashion. Additionally, we introduce a cost aggregation module based on

---

*part of the work was done during an internship at Microsoft Research Asia

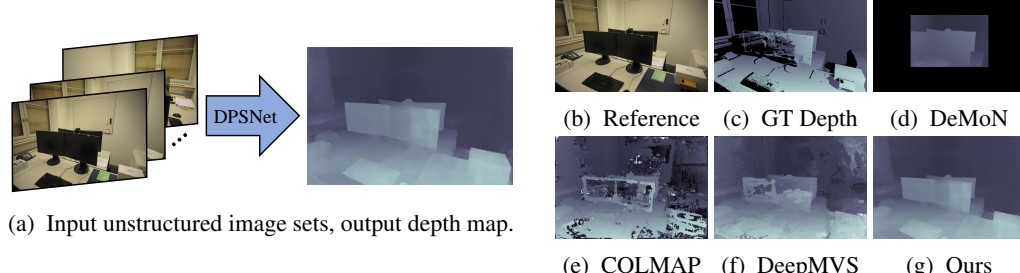

(a) Input unstructured image sets, output depth map.

(b) Reference    (c) GT Depth    (d) DeMoN

(e) COLMAP    (f) DeepMVS    (g) Ours

Figure 1: Results of DPSNet with comparisons to state-of-the-art methods.

local cost-volume filtering Rhemann et al. (2011) for context-aware refinement of each cost slice. Through this cost-volume regularization, the effects of unreliable matches scattered within the cost volume are reduced considerably.

With this end-to-end network for plane-sweep stereo and the proposed cost aggregation, we obtain state-of-the-art results over several standard datasets. Ablation studies indicate that each of these technical contributions leads to appreciable improvements in reconstruction accuracy.

## 2 RELATED WORK

CNN-based depth estimation has been studied for stereo matching, depth from single images, and multiview stereo. Recent work in these areas are briefly reviewed in the following.

**Stereo matching** Methods for stereo matching address the particular case of depth estimation where the input is a pair of rectified images captured by a stereo rig. Various network structures have been introduced for this problem. Zbontar & LeCun (2016) present a Siamese network structure to compute matching costs based on the similarity of two image patches. The estimated initial depth is then refined by traditional cost aggregation and refinement as post-processing. Mayer et al. (2016) directly stack several convolution and deconvolution layers upon the matching costs and train the network to minimize the distance between the estimates and ground truth. Liang et al. (2018) propose a CNN that estimates initial disparity and then refines it using both prior and posterior feature consistency in an end-to-end manner. Kendall et al. (2017) leverage geometric knowledge in building a cost volume from deep feature representations. It also enables learning of contextual information in a 3D volume and regresses disparity in an end-to-end manner. Chang & Chen (2018) introduce a pyramid pooling module for incorporating global contextual information into image features and a stacked hourglass 3D CNN to extend the regional support of contextual information.

**Depth from single images** Similar to these stereo matching approaches, single-image methods extract CNN features to infer scene depths and perform refinements to increase depth accuracy. The first of these methods was introduced by Eigen et al. (2014), which demonstrated that CNN features could be utilized for depth inference. Later, Liu et al. (2016) combined a superpixel-based conditional random field (CRF) to a CNN to improve the quality of depth estimates from single images. To facilitate training, recent studies Zhou et al. (2017); Wang et al. (2018); Mahjourian et al. (2018); Yin & Shi (2018) present an end-to-end learning pipeline that utilizes the task of view synthesis as supervision for single-view depth and camera pose estimation. These systems consist of a depth network and a pose estimation network which simultaneously train on sequential images with a loss computed from images warped to nearby views using the estimated depth. View synthesis has similarly been used as supervision by warping between stereo image pairs Garg et al. (2016); Godard et al. (2017). In contrast to these single-image works which employ warping as a component of view synthesis for self-supervised learning, our network computes warps with respect to multiple depth planes to produce plane-sweep cost volumes both for training and at test time. The cost volumes undergo further processing in the form of cost aggregation and regularization to improve the robustness of depth estimates.

**Multi-view stereo** In multi-view stereo, depth is inferred from multiple input images acquired from arbitrary viewpoints. To solve this problem, some methods recover camera motion between

the unstructured images but are designed to handle only two views Ummenhofer et al. (2017); Li et al. (2018). The DeMoN system Ummenhofer et al. (2017) consists of encoder-decoder networks for optical flow, depth/motion estimation, and depth refinement. By alternating between estimating optical flow and depth/motion, the network is forced to use both images in estimating depth, rather than resorting to single-image inference. Li et al. (2018) perform monocular visual odometry in an unsupervised manner. In the training step, the use of stereo images with extrinsic parameters allows 3D depth estimation to be estimated with metric scale.

Among networks that can handle an arbitrary number of views, camera parameters are assumed to be known or estimated by conventional geometric methods. Ji et al. (2017) introduce an end-to-end learning framework based on a viewpoint-dependent voxel representation which implicitly encodes images and camera parameters. The voxel representation restricts the scene resolution that can be processed in practice due to limitations in GPU memory. Im et al. (2018b) formulate a geometric relationship between optical flow and depth to refine the estimated scene geometry, but is designed for image sequences with a very small baseline, i.e., an image burst from a handheld camera. Huang et al. (2018) compute a set of plane-sweep volumes using calibrated pose data as input for the network, which then predicts an initial depth feature using an encoder-decoder network. In the depth prediction step, they concatenate a reference image feature to the decoder input as an intra-feature aggregation, and cost volumes from each of the input images are aggregated by max-pooling to gather information for the multiview matching. Its estimated depth map is refined using a conventional CRF. By contrast, our proposed DPSNet is developed to be trained end-to-end from input images to the depth map. Moreover, it leverages conventional multiview stereo concepts by incorporating context-aware cost aggregation. Finally, we would like to refer the reader to the concurrent work by Yao et al. (2018) that also adopts differential warping to construct a multi-scale cost volume, then refined an initial depth map guided by a reference image feature. Our work is independent of this concurrent effort. Moreover, we make distinct contributions: (1) We focus on dense depth estimation for a reference image in an end-to-end learning manner, different from Yao et al. (2018) which reconstructs the full 3D of objects. (2) Our cost volume is constructed by concatenating input feature maps, which enables inference of accurate depth maps even with only two-view matching. (3) Our work refines every cost slice by applying context features of a reference image, which is beneficial for alleviating coarsely scattered unreliable matches such as for large textureless regions.

## 3 Approach

Our Deep Plane Sweep Network (DPSNet) is inspired by traditional multiview stereo practices for dense depth estimation and consists of four parts: feature extraction, cost volume generation, cost aggregation and depth map regression. The overall framework is shown in Figure 2.

### 3.1 Multi-scale Feature Extraction

We first pass a reference image and target images through seven convolutional layers ($3 \times 3$ filters except for the first layer, which has a $7 \times 7$ filter) to encode them, and extract hierarchical contextual information from these images using a *spatial pyramid pooling* (SPP) module He et al. (2014) with four fixed-size average pooling blocks ($16 \times 16, 8 \times 8, 4 \times 4, 2 \times 2$). The multi-scale features extracted by SPP have been shown to be effective in many visual perception tasks such as visual recognition He et al. (2014), scene parsing Zhao et al. (2017) and stereo matching Huang et al. (2018). After upsampling the hierarchical contextual information to the same size as the original feature map, we concatenate all the feature maps and pass them through 2D convolutional layers. This process yields 32-channel feature representations for all the input images, which are next used in building cost volumes.

### 3.2 Cost Volume Generation using Unstructured Two-View Images

We propose to generate cost volumes for the multiview images by adopting traditional plane sweep stereo Collins (1996); Yang & Pollefeys (2003); Ha et al. (2016); Im et al. (2018a), originally devised for dense depth estimation. The basic idea of plane sweep stereo is to back-project the image set onto successive virtual planes in the 3D space and measure photo-consistency among the warped

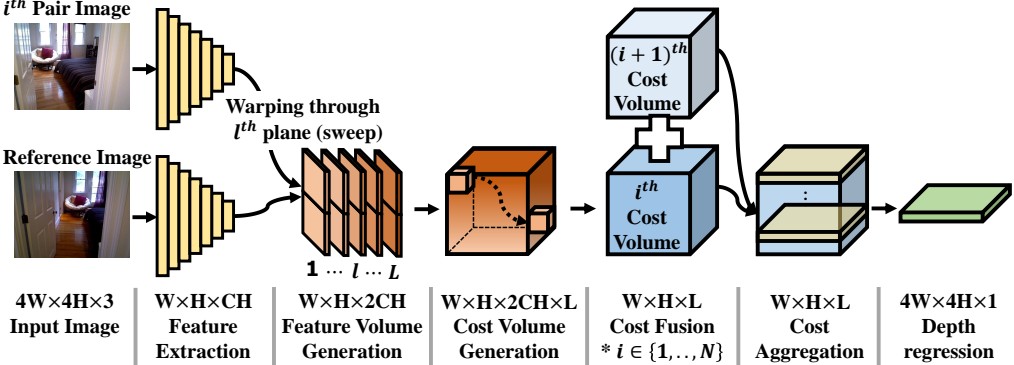

Figure 2: Overview of the DPSNet pipeline.

images for each pixel. In a similar manner to traditional plane sweep stereo, we construct a cost volume from an input image pair. To reduce the effects of image noise, multiple images can be utilized by averaging cost volumes for other pairs.

For this cost volume generation network, we first set the number of virtual planes perpendicular to the $z$-axis of the reference viewpoint $[0, 0, 1]^\intercal$ and uniformly sample them in the inverse-depth space as follows:

$$d_l = \frac{(L \times d_{\min})}{l}, \ (l = 1, .., L), \tag{1}$$

where $L$ is the total number of depth labels and $d_{\min}$ is the minimum scene depth as specified by the user.

Then, we warp all the paired features $\mathcal{F}_i$, $(i = 1, .., N)$, where $i$ is an index of viewpoints and $N$ is the total number of input views, into the coordinates of the reference feature (of size $Width \times Height \times CHannel$) using pre-computed intrinsics $\mathbf{K}$ and extrinsic parameters consisting of a rotation matrix $\mathbf{R}_i$ and a translation matrix $\mathbf{t}_i$ of the $i^{th}$ camera:

$$\tilde{\mathcal{F}}_{il}(u) = \mathcal{F}_i(\tilde{u}_l), \ \ \tilde{u}_l \sim \mathbf{K}[\mathbf{R}_i|\mathbf{t}_i] \begin{bmatrix} (\mathbf{K}^{-1}u)d_l \\ 1 \end{bmatrix}, \tag{2}$$

where $u, \tilde{u}_l$ are the homogeneous coordinates of a pixel in the reference view and the projected coordinates onto the paired view, respectively. $\tilde{\mathcal{F}}_{il}(u)$ denotes the warped features of the paired image through the $l^{th}$ virtual plane. Unlike the traditional plane sweeping method which utilizes a distance metric, we use a concatenation of features in learning a representation and carry this through to the cost volume as proposed in Kendall et al. (2017). We obtain a 4D volume ($W \times H \times 2CH \times L$) by concatenating the reference image features and the warped image features for all of the depth labels. In Eq. (2), we assume that all images are captured by the same camera, but it can be directly extended to images with different intrinsics. For the warping process, we use a *spatial transformer network* Jaderberg et al. (2015) for all hypothesis planes, which does not require any learnable parameters. In Table 3, we find that concatenating features improves performance over the absolute difference of the features.

Given the 4D volume[1], our DPSNet learns a cost volume generation of size $W \times H \times L$ by using a series of 3D convolutions on the concatenated features. All of the convolutional layers consist of $3 \times 3 \times 3$ filters and residual blocks. In the training step, we only use one paired image (while the other is the reference image) to obtain the cost volume. In the testing step, we can use any number of paired images ($N \geq 1$) by averaging all of the cost volumes.

### 3.3 COST AGGREGATION

The key idea of cost aggregation Rhemann et al. (2011) is to regularize the noisy cost volume through edge-preserving filtering He et al. (2013) within a support window. Inspired by traditional cost

---

[1]We implement a 5D volume that includes a batch dimension.

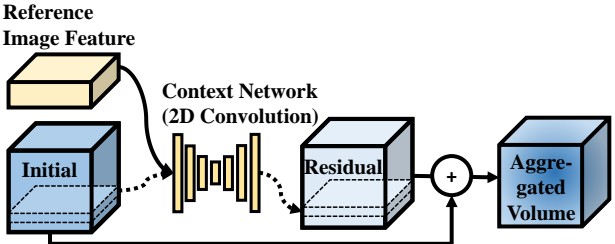

Figure 3: Illustration of context-aware cost aggregation.

volume filtering, we introduce a context-aware cost aggregation method in our end-to-end learning process. The context network takes each slice of the cost volume and the reference image features extracted from the previous step, and then outputs the refined cost slice. We run the same process for all the cost slices. The final cost volume is then obtained by adding the initial and residual volumes as shown in Figure 3.

Here, we use dilated convolutions in the context network for cost aggregation to better exploit contextual information Chen et al. (2018); Yu & Koltun (2016). The context network consists of seven convolutional layers with $3 \times 3$ filters, where each layer has a different receptive field (1, 2, 4, 8, 16, 1, and 1). We jointly learn all the parameters, including those of the context network. All cost slices are processed with shared weights of the context network. Then, we upsample the cost volume, whose size is equal to the feature size, to the original size of the images via bilinear interpolation. We find that this leads to moderate performance improvement as shown in Table 3.

## 3.4 DEPTH REGRESSION

We regress continuous depth values using the method proposed in Kendall et al. (2017). The probability of each label $l$ is calculated from the predicted cost $c_l$ via the softmax operation $\sigma(\cdot)$. The predicted label $\hat{l}$ is computed as the sum of each label $l$ weighted by its probability. With the predicted label, the depth is calculated from the number of labels $L$ and minimum scene depth $d_{\min}$ as follows:

$$\tilde{d} = \frac{L \times d_{\min}}{\tilde{l}}, \quad \tilde{l} = \sum_{l=1}^{L} l \times \sigma(c_l). \tag{3}$$

We set $L$ and $d_{\min}$ to 64 and 0.5, respectively.

## 3.5 TRAINING LOSS

Let $\theta$ be the set of all the learnable parameters in our network, which includes feature extraction, cost volume generation and cost aggregation (plane sweep and depth regression have no learnable parameters). Let $\hat{d}, \tilde{d}$ denote the predicted depth from the initial and refined cost volumes, respectively, and let $d^{gt}$ be the corresponding supervision signal. The training loss is then formulated as

$$\mathcal{L}(\theta) = \sum_{\mathbf{x}} \lambda |\hat{d}_{\mathbf{x}}^{\theta} - d_{\mathbf{x}}^{gt}|_{\mathbf{H}} + |\tilde{d}_{\mathbf{x}}^{\theta} - d_{\mathbf{x}}^{gt}|_{\mathbf{H}}, \tag{4}$$

where $|\cdot|_{\mathbf{H}}$ denotes the Huber norm, referred to as *SmoothL1Loss* in PyTorch. The weight value $\lambda$ for depth from the initial cost volume is set to 0.7.

| Data -sets | Method | Error metric | | | | | Accuracy metric ($\delta < \alpha^t$) | | |
|---|---|---|---|---|---|---|---|---|---|
| | | Abs Rel | Abs diff | Sq Rel | RMSE | RMSE log | $\alpha$ | $\alpha^2$ | $\alpha^3$ |
| MVS | COLMAP | 0.3841 | 0.8430 | 1.257 | 1.4795 | 0.5001 | 0.4819 | 0.6633 | 0.8401 |
| | DeMoN | 0.3105 | 1.3291 | 19.970 | 2.6065 | 0.2469 | 0.6411 | 0.9017 | 0.9667 |
| | DeepMVS | 0.2305 | 0.6628 | 0.6151 | 1.1488 | 0.3019 | 0.6737 | 0.8867 | 0.9414 |
| | Ours | **0.0722** | **0.2095** | **0.0798** | **0.4928** | **0.1527** | **0.8930** | **0.9502** | **0.9760** |
| SUN3D | COLMAP | 0.6232 | 1.3267 | 3.2359 | 2.3162 | 0.6612 | 0.3266 | 0.5541 | 0.7180 |
| | DeMoN | 0.2137 | 2.1477 | 1.1202 | 2.4212 | 0.2060 | 0.7332 | 0.9219 | 0.9626 |
| | DeepMVS | 0.2816 | 0.6040 | 0.4350 | 0.9436 | 0.3633 | 0.5622 | 0.7388 | 0.8951 |
| | Ours | **0.1470** | **0.3234** | **0.1071** | **0.4269** | **0.1906** | **0.7892** | **0.9317** | **0.9672** |
| RGBD | COLMAP | 0.5389 | 0.9398 | 1.7608 | 1.5051 | 0.7151 | 0.2749 | 0.5001 | 0.7241 |
| | DeMoN | 0.1569 | 1.3525 | 0.5238 | 1.7798 | **0.2018** | 0.8011 | 0.9056 | 0.9621 |
| | DeepMVS | 0.2938 | 0.6207 | 0.4297 | 0.8684 | 0.3506 | 0.5493 | 0.8052 | 0.9217 |
| | Ours | **0.1538** | **0.5235** | **0.2149** | **0.7226** | 0.2263 | **0.7842** | **0.8959** | **0.9402** |
| Scenes11 | COLMAP | 0.6249 | 2.2409 | 3.7148 | 3.6575 | 0.8680 | 0.3897 | 0.5674 | 0.6716 |
| | DeMoN | 0.5560 | 1.9877 | 3.4020 | 2.6034 | 0.3909 | 0.4963 | 0.7258 | 0.8263 |
| | DeepMVS | 0.2100 | 0.5967 | 0.3727 | 0.8909 | 0.2699 | 0.6881 | 0.8940 | 0.9687 |
| | Ours | **0.0558** | **0.2430** | **0.1435** | **0.7136** | **0.1396** | **0.9502** | **0.9726** | **0.9804** |

Table 1: **Comparison results.** Multi-view stereo methods: COLMAP, DeMoN, DeepMVS, and Ours. Datasets: MVS, SUN3D, RGBD, and Scenes11.

| Method | Compl -eteness | Error metric | | | | | | | Accuracy metric ($\delta < \alpha^t$) | | |
|---|---|---|---|---|---|---|---|---|---|---|---|
| | | Geo. | Photo. | A. Rel | A. diff | Sq Rel | RMSE | Rlog | $\alpha$ | $\alpha^2$ | $\alpha^3$ |
| COLMAP filter | 71 % | 0.007 | 0.178 | 0.045 | 0.033 | **0.293** | 0.619 | 0.123 | 0.965 | 0.978 | 0.986 |
| COLMAP | 100 % | 0.046 | 0.218 | 0.324 | 0.615 | 36.71 | 2.370 | 0.349 | **0.865** | 0.903 | 0.927 |
| DeMoN | 100 % | 0.045 | 0.288 | 0.191 | 0.726 | 0.365 | 1.059 | 0.240 | 0.733 | 0.898 | 0.951 |
| DeepMVS | 100 % | 0.036 | 0.224 | 0.178 | 0.432 | 0.973 | 1.021 | 0.245 | 0.858 | 0.911 | 0.942 |
| MVSNET filter | 77 % | 0.067 | **0.179** | 0.357 | 0.766 | 1.969 | 1.325 | 0.423 | 0.706 | 0.779 | 0.829 |
| MVSNET | 100 % | 0.077 | 0.218 | 1.666 | 2.165 | 13.93 | 3.255 | 0.824 | 0.555 | 0.628 | 0.686 |
| Ours | 100 % | **0.034** | 0.202 | **0.099** | **0.365** | 0.204 | **0.703** | **0.184** | 0.863 | **0.938** | **0.963** |

Table 2: **Comparison results.** Multi-view stereo methods on ETH3D. The 'filter' refers to predicted disparity maps from outlier rejection. (Geo, Photo: Geometric and Photometric error; A. Rel: Abs rel; A. diff: Abs diff; Rlog: RMSE log.) Underbar: Best, **Bold**: Second best.

## 4 EXPERIMENTS

### 4.1 IMPLEMENTATION DETAILS

In the training procedure, we use image sequences, ground-truth depth maps for reference images, and the provided camera poses from public datasets, namely SUN3D, RGBD, and Scenes11[2]. We train our model from scratch for 1200K iterations in total. All models were trained end-to-end with the ADAM optimizer ($\beta_1 = 0.9$, $\beta_2 = 0.999$). We use a batch size of 16 and set the learning rate to $2e-4$ for all iterations. The training is performed with a customized version of PyTorch on four NVIDIA 1080Ti GPUs, which usually takes four days. A forward pass of the proposed network takes about 0.5 seconds for 2-view matching and an additional 0.25 seconds for every new frame matched ($640 \times 480$ image resolution).

### 4.2 COMPARISON WITH STATE-OF-THE-ART METHODS

In our evaluations, we use common quantitative measures of depth quality: absolute relative error (Abs Rel), absolute relative inverse error (Abs R-Inv), absolute difference error (Abs diff), square relative error (Sq Rel), root mean square error and its log scale (RMSE and RMSE log) and inlier ratios ($\delta < 1.25^i$ where $i \in \{1, 2, 3\}$). All are standard metrics used in a public benchmark suite[3].

For our comparisons, we choose state-of-the-art methods for traditional geometry-based multiview stereo (COLMAP) Schönberger & Frahm (2016), depth from unstructured two-view stereo (DeMoN) Ummenhofer et al. (2017) and CNN-based multiview stereo (DeepMVS) Huang et al. (2018). We estimate the depth maps from two unstructured views using the test sets in MVS, SUN3D, RGBD and Scenes11, as done for DeMoN[4].

---

[2]https://github.com/lmb-freiburg/demon
[3]http://www.cvlibs.net/datasets/kitti/
[4]We use the provided camera intrinsics and extrinsics.

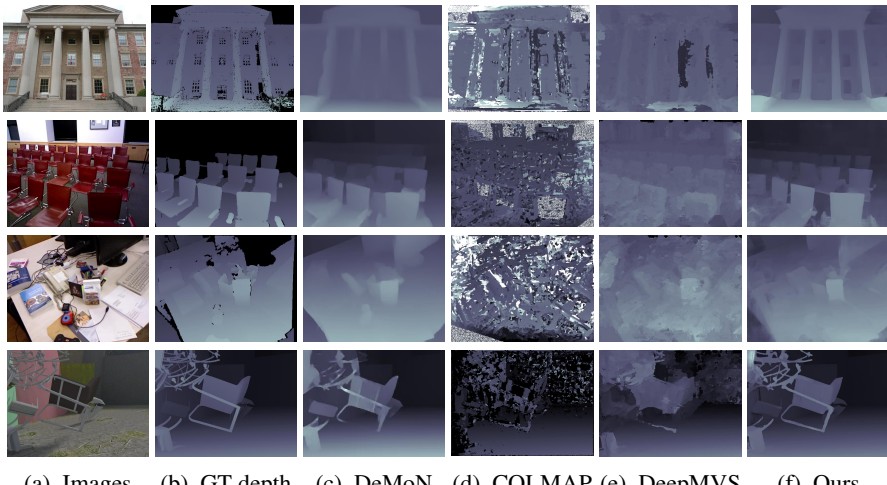

   (a) Images    (b) GT depth   (c) DeMoN   (d) COLMAP  (e) DeepMVS   (f) Ours

Figure 4: Comparison of depth map results on MVS, SUN3D, RGBD and Scenes11 (top to bottom).

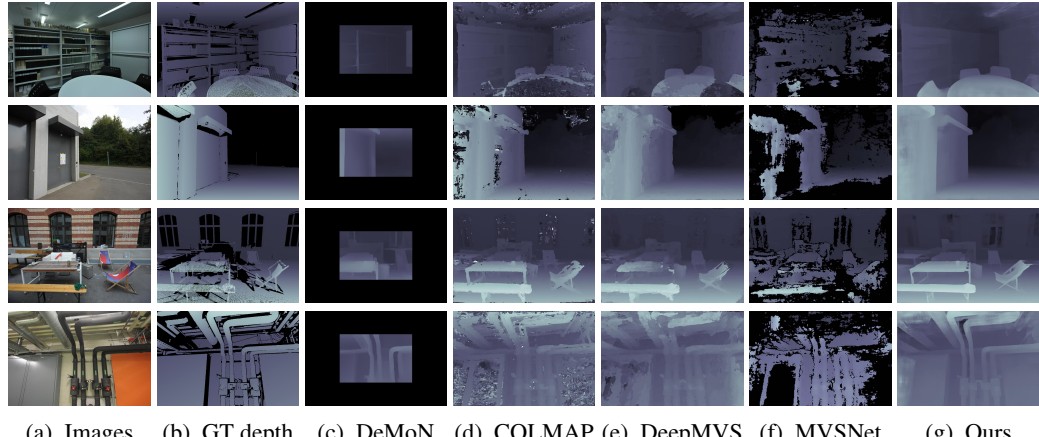

  (a) Images   (b) GT depth   (c) DeMoN   (d) COLMAP  (e) DeepMVS  (f) MVSNet   (g) Ours

Figure 5: Depth map results on four-view image sequences from the ETH3D dataset.

The results are reported in Table 1. Our DPSNet provides the best performance on nearly all of the measures. Of particular note, DPSNet accurately recovers scene depth in homogeneous regions as well as along object boundaries as exhibited in Figure 4. DeMoN generally produces good depth estimates but often fails to reconstruct scene details such as the keyboard (third row) and fine structures (first, second and fourth rows). By contrast, DPSNet estimates accurate depth maps at those regions because the differential feature warping penalizes inaccurate reconstructions, playing a role similar to the left-right consistency check that has been used in stereo matching Garg et al. (2016). The first and third rows of Figure 4 exhibit problems of COLMAP and DeepMVS in handling textureless regions. DPSNet instead produces accurate results, courtesy of the cost aggregation network.

For a more balanced comparison, we adopt measures used in Huang et al. (2018) as additional evaluation criteria: (1) completeness, which is the percentage of pixels whose errors are below a certain threshold. (2) geometry error, taking the L1 distance between the estimated disparity and the ground truth. (3) photometry error, which is the L1 distance between the reference image and warped image using the estimated disparity map. The results for COLMAP, DeMoN and DeepMVS are directly reported from Huang et al. (2018) in Table 2. In this experiment, we use the ETH3D dataset on which all methods are not trained. Following Yao et al. (2018), we take 5 images with $1152 \times 864$ resolution and set 192 depth labels based on ground-truth depth to obtain optimal results for MVSNet. For the DPSNet results, we use 4 views with $810 \times 540$ resolution and set 64 labels whose range is determined by the minimum depth values of the ground truth.

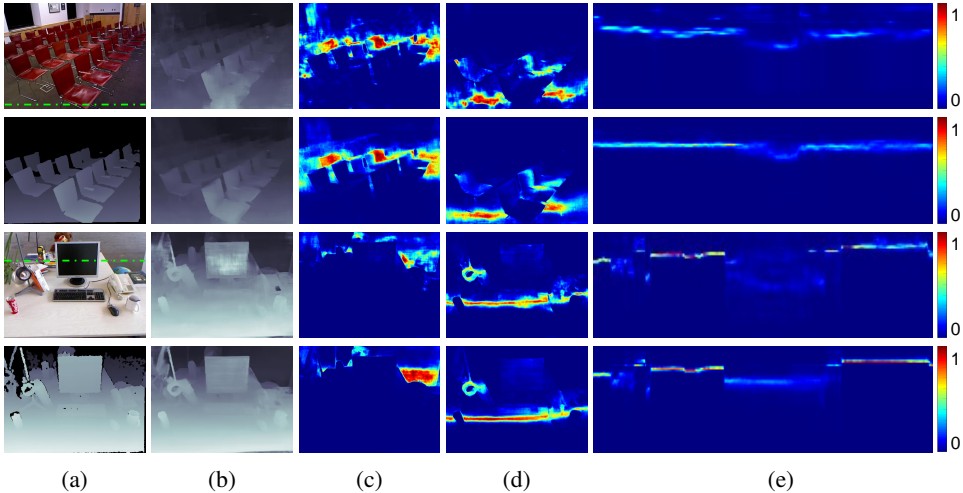

|  | (a) | (b) | (c) | (d) | (e) |

Figure 6: (a) Reference image & GT depth. (b) Regressed depth. (c), (d) Slice of volume along a label (far & Close), and (e) along the green row in (a) (Column-Cost layer axis). The color bar ranges from 0 to 1. Before (top) and after (bottom) cost volume aggregation.

| Method | Error metric | | | | | Accuracy metric | | |
|---|---|---|---|---|---|---|---|---|
|  | Abs Rel | Abs diff | Sq Rel | RMSE | RMSE log | $\delta < 1.25$ | $\delta < 1.25^2$ | $\delta < 1.25^3$ |
| (a) Cost (Difference) | 0.1096 | 0.3006 | 0.1508 | 0.5484 | 0.1780 | 0.8643 | 0.9442 | 0.9735 |
| (b) W/O Aggregation | 0.1274 | 0.3388 | 0.1957 | 0.6230 | 0.2112 | 0.8372 | 0.9268 | 0.9616 |
| (c) Stacked hourglass | **0.0994** | 0.2864 | 0.1377 | 0.5306 | 0.1734 | **0.8774** | 0.9494 | 0.9734 |
| (d) Uniform depth | 0.1272 | 0.3429 | 0.1758 | 0.5951 | 0.2099 | 0.8381 | 0.9235 | 0.9564 |
| (e) Ours | 0.1028 | **0.2852** | **0.1352** | **0.5207** | **0.1704** | 0.8733 | **0.9495** | **0.9759** |

Table 3: **Ablation Experiment.** (a) With cost volume generated by absolute differences of features. (b) Without cost aggregation. (c) With cost aggregation by stacked hourglass. (d) With planes swept on uniform samples of the depth domain from $0.5m$ to $10m$ (whereas ours are uniformly sampled on the inverse depth domain from $0.5m$). (e) With our complete DPSNet. Datasets: MVS, SUN3D, RGBD, Scenes11. Note that we masked out the depth beyond the range $[0.5, 10]$ for the evaluation.

In Table 2, our DPSNet shows the best performance overall among the all the comparison methods, except for filtered COLMAP. Although filtered COLMAP achieves the best performance, its completeness is only 71% and its unfiltered version shows a significant performance drop in all error metrics. On the other hand, our DPSNet with 100% completeness shows promising results on all measures. We note that our DPSNet has a different purpose compared to COLMAP and MVSNet. COLMAP and MVSNet are designed for full 3D reconstruction with an effective outlier rejection process, while DPSNet aims to estimate a dense depth map for a reference view.

## 4.3 ABLATION STUDY

An extensive ablation study was conducted to examine the effects of different components on DP-SNet performance. We summarize the results in Table 3.

**Cost Volume Generation** In Table 3 (a) and (e), we compare the use of cost volumes generated using the traditional absolute difference Collins (1996) and using the concatenation of features from the reference image and warped image. The absolute difference is widely used for depth label selection via a winner-take-all strategy. However, we observe that feature concatenation provides better performance in our network than the absolute difference. A possible reason is that the CNN may learn to extract 3D scene information from the tensor of stacked features. The tensor is fed into the CNN to produce an effective feature for depth estimation, which is then passed through our cost aggregation network for the initial depth refinement.

**Cost Aggregation** For our cost aggregation sub-network, we compare DPSNet with and without it in Table 3 (e) and (b), respectively. It is shown that including the proposed cost aggregation

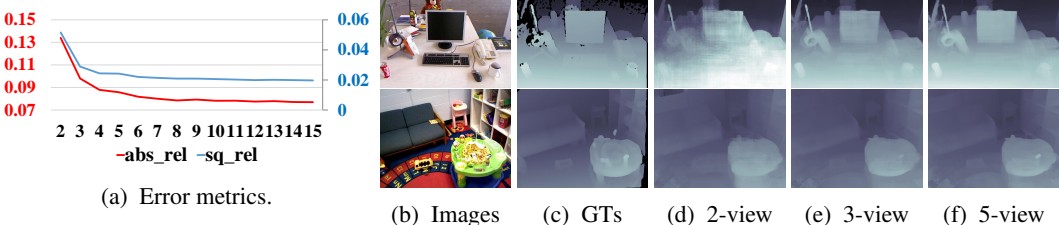

(a) Error metrics.   (b) Images   (c) GTs   (d) 2-view   (e) 3-view   (f) 5-view

Figure 7: Depth map results w.r.t. the number of images.

leads to significant performance improvements. Examples of depth map refinement with the cost aggregation are displayed in Figure 6.

Our cost aggregation is also compared to using a stacked hourglass to aggregate feature information along the depth dimension as well as the spatial dimensions as done recently for stereo matching Chang & Chen (2018). Although the stacked hourglass is shown in Table 3 (c) to enhance depth results, its improvement is smaller than ours, which uses reference image features to guide the aggregation. Figure 6 (a), (b) show examples of the depth map results before and after our cost aggregation. It demonstrates that the cost aggregation sub-network outputs more accurate depth, especially on homogeneous regions.

|  | Winner Margin | Curvature |
|---|---|---|
| Before Aggregation | 0.7001 | 0.4614 |
| After Aggregation | **0.7136** | **0.4836** |

Table 4: Confidence measures on cost volumes. Datasets: MVS, SUN3D, RGBD, Scenes11.

For further analysis of cost aggregation, we display slices of 3D cost volumes after the softmax operation (in Eq. (3)) that span depth labels and the rows of the images. The cost slices in Figure 6 (c), (d) show that our feature-guided cost aggregation regularizes noisy cost slices while preserving edges well. The cleaner cost profiles that ensue from the cost aggregation lead to clearer and edge-preserving depth regression results. As mentioned in a recent study Hu & Mordohai (2012), a cost profile that gives confident estimates should have a single, distinct minimum (or maximum), while an ambiguous profile has multiple local minima or multiple adjacent labels with similar costs, making it hard to exactly localize the global minimum. Based on two quantitative confidence measures Hu & Mordohai (2012) on cost volumes in Table 4, the proposed aggregation improves the reliability of the correct match corresponding to the minimum cost.

**Depth Label Sampling** In the plane sweep procedure, depth labels can be sampled in either the depth domain or the inverse depth domain, which provides denser sampling in areas closer to a camera. Table 3 (d) and (e) show that uniform depth label sampling in the inverse depth domain produces more accurate depth maps in general.

**Number of Images** We examine the performance of DPSNet with respect to the number of input images. As displayed in Figure 7a, a greater number of images yields better results, since cost volume noise is reduced through averaging over more images, and more viewpoints help to provide features from areas unseen in other views. Figure 7 shows that adding input views aids in distinguishing object boundaries. Note that the performance improvement plateaus when seven or more images are used.

**Rectified Stereo Pair** CNNs-based stereo matching methods have similarity to DPSNet, but differ from it in that correspondences are obtained by shifting learned features in Mayer et al. (2016); Kendall et al. (2017); Tulyakov et al. (2018). The purpose of this study is to show readers that not only descriptor shift but also plane sweeping can be applied to rectified stereo matching. We apply DPSNet on the KITTI dataset, which provides rectified stereo pairs with a specific baseline.

As shown in Figure 8, although DPSNet is not designed to work on rectified stereo images, it produces reasonable results. In particular, DPSNet fine-tuned on the KITTI dataset in Table 5 achieves performance similar to Mayer et al. (2016) in terms of D1-all score, with 4.34% for all pixels and 4.05% for non-occluded pixels in the KITTI benchmark. We expect that the depth accuracy would

| W/O ft | D1-bg | D1-fg | D1-all | With ft | D1-bg | D1-fg | D1-all |
|--------|-------|-------|--------|---------|-------|-------|--------|
| All / All | 4.69 % | 17.77 % | 6.87 % | All / All | 4.21 % | 7.58 % | 4.77 % |
| Noc / All | 4.23 % | 16.30 % | 6.23 % | Noc / All | 3.58 % | 6.08 % | 4.00 % |

Table 5: KITTI2015 Benchmark without/with finetuning. D1 error denotes the percentage of stereo disparity outliers in the first frame.

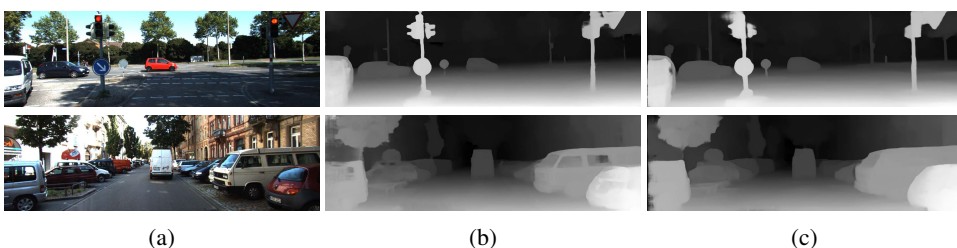

(a)                (b)                (c)

Figure 8: Rectified stereo evaluation. (a) Reference images. (b) Depth map results without fine-tuning. (c) Depth map results with fine-tuning. (KITTI2015 test dataset).

improve if we were to adopt rectified stereo pair-specific strategies, such as the feature consistency check in Liang et al. (2018).

## 5 DISCUSSION

We developed a multiview stereo network whose design is inspired by best practices of traditional non-learning-based techniques. The plane sweep algorithm is formulated as an end-to-end network via a differentiable construction of plane sweep cost volumes and by solving for depth as a multi-label classification problem. Moreover, we propose a context-aware cost aggregation method that leads to improved depth regression without any post-processing. With this incorporation of traditional multiview stereo schemes into a deep learning framework, state-of-the-art reconstruction results are achieved on a variety of datasets.

Directions exist for improving DPSNet. One is to integrate semantic instance segmentation into the cost aggregation, similar to the segment-based cost aggregation method of Mei et al. (2013). Another direction is to improve depth prediction by employing viewpoint selection in constructing cost volumes Gallup et al. (2008); Schönberger et al. (2016), rather than by simply averaging the estimated cost volumes as currently done in DPSNet. Lastly, the proposed network requires pre-calibrated intrinsic and extrinsic parameters for reconstruction. Lifting this restriction by additionally estimating camera poses in an end-to-end learning framework is an important future challenge.

## ACKNOWLEDGEMENT

This work was supported by the Technology Innovation Program (No. 2017-10069072) funded By the Ministry of Trade, Industry & Energy (MOTIE, Korea), and supported in part by Air Force Research Laboratory (AFRL) project FA23861714660. Sunghoon Im was partially supported by Global Ph.D. Fellowship Program through the National Research Foundation of Korea (NRF) funded by the Ministry of Education (NRF-2016907531). Hae-Gon Jeon was partially supported by Basic Science Research Program through the National Research Foundation of Korea (NRF) funded by the Ministry of Education (2018R1A6A3A03012899).

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
