# OpenReview forum: "DPSNet: End-to-end Deep Plane Sweep Stereo"
_ICLR.cc/2019/Conference_

### Official Review · AnonReviewer3 · 2018-10-30
**Works well, maybe too straightforward for ICLR**

**Rating:** 6
**Confidence:** 4

**Review:**

This paper proposes a method for stereo reconstruction using Deep Learning. Like some previous methods, a 'cost volume' is first computed by plane sweeping, in other words the cost volume is indexed by the 2D locations in the image plane, and the disparities for 3D planes parallel to the image plane. A network then predicts the disparities for each image location from this cost volume.

The contributions with respect to the state-of-the-art are:

- the cost volume is computed using differential warps, thus the network can be trained end-to-end;

- a better cost volume is computed from the original cost volume and the reference image.

The results look good, both quantitatively and qualitatively. The paper reads well, and related work is correctly referenced.

There is nothing wrong with the proposed method, it makes sense and I am convinced it works well. However, I found the contributions quite straightforward, and it is difficult to get excited about the paper.

More details would be welcome for Section 3.2

---

> ### Author Response · Authors · 2018-11-21
> **Response to Reviewer3**
>
> More details would be welcome for Section 3.2
> Thanks for your positive comments on our paper. Since your comment is the same as that of Reviewer 1, please refer to the response 1 to Reviewer 1’s question and check our revised manuscript.

---

### Official Review · AnonReviewer2 · 2018-11-01
**Practically good but technically weak.**

**Rating:** 6
**Confidence:** 5

**Review:**

Summary
This paper proposes an end-to-end learnable  multiview stereo depth estimation network, which is basically very similar to the GCNet (Kendall et.al 2017) or PSMNet (Chang et.al 2018) for stereo estimation. The differences are using SPN to warp feature w.r.t RT, adding a multi view averaging cost and a cost aggregation component for final depth regression, which transform the original network to support multi-view stereo, yielding performance boost over other baselines.

Technically, I believe it is sound  because cost volume from stereo matching has already been demonstrate very effective in boosting performance because it use underlining geometry constraint.  My major concern lies in three aspects.

1) Another most recent SOTA algorithm is  MVSNet (Yao et.al ECCV 2018), the paper should be considered for comparison. In addition, the structure is even more similar with the proposed network architecture.

2) The evaluation metrics are mostly use for single view depths, it is not consistent with paper of DeepMVS (Tab. 1) or that from MVSNet. Therefore, it might be hard to actual understand whether the numbers are  exactly comparable.

3) Since the method largely improved over their baseline algorithms, and the number between different papers are hard to compare. In my opinion, to better show the results,  I suggest submitting results to an online benchmark with test data for verifying the results. such as ETH3D multi view benchmark, where everything is standardized.

I hope the author can make strong feedback for validating the results.

####### . After rebuttal

The author makes more clear indication of the performance contribution of the completeness of recovery.

---

> ### Author Response · Authors · 2018-11-21
> **Response to Reviewer2 (a)**
>
> 1) Another most recent SOTA algorithm is MVSNet (Yao et.al ECCV 2018), the paper should be considered for comparison. In addition, the structure is even more similar with the proposed network architecture.
> Thanks for your suggestion to improve our experiments. Following your comment, we have added a description of MVSNet at the end of Sec. 2 and mentioned the differences from our DPSNet. Traditional stereo matching and multiview stereo consist of four steps: initial matching, cost aggregation, multi-label optimization and refinement. In our opinion, MVSNet mainly focuses on computing initial matching and refinement. Our contributions are mainly on initial matching and cost aggregation. Even though the main scheme for warping is similar, the cost volume generation and aggregation approaches are different.
>
> MVSNet was developed concurrently to our work, and there are differences in contributions. In particular, (1) our DPSNet concatenates the reference image features and warped pair image features, then builds a cost volume. If multiple views are available (more than 3 views), we iteratively compute the cost volume and average them. This cost volume generation strategy can produce a more confident cost volume as more views are matched. Moreover, we show that our DPSNet can be generalized to binocular stereo, while MVSNet builds a cost volume based on the variance of features, which fails to estimate accurate depth values when only using two views. (2) Our novel cost aggregation network shows significant performance improvements on textureless regions over MVSNet’s refinement with a reference image feature. (3) Both the minimum and maximum depth range are user-defined parameters and scene-dependent. As an alternative, our DPSNet uniformly samples matching planes in inverse-depth scale, which is beneficial for alleviating depth quantization error and reconstructing scenes with large depth ranges.
> We cannot say with certainty which method performs better. What we can say clearly is that both methods have shown great performance improvements by applying traditional techniques used in multiview stereo into deep learning architectures. In other words, both studies have independent academic contributions.
> For a comparison to MVSNet, please refer to the next response.

---

> ### Author Response · Authors · 2018-11-21
> **Response to Reviewer2 (b)**
>
> 2) The evaluation metrics are mostly use for single view depths, it is not consistent with paper of DeepMVS (Tab. 1) or that from MVSNet. Therefore, it might be hard to actual understand whether the numbers are exactly comparable.
> The error metrics used in this paper are for dense depth measurements on the KITTI benchmark. Since the main goal of our work is to reconstruct a dense depth map from multiple images, we think the error metrics are suitable for performance evaluation of our DPSNet and the state-of-the-art methods. However, we agree with your view and report a quantitative evaluation using the three error metrics used in DeepMVS: (1) Completeness, which is the percentage of pixels whose errors are below a certain threshold. (2) Geometry error, taking the L1 distance between the estimated disparity and the ground truth. (3) Photometry error, which is the L1 distance between the reference image and warped image using the estimated disparity map.
>
> We evaluated depth maps from our DPSNet and all comparison methods including MVSNet using those measures. For fair comparison, we use the ETH3D dataset on which all methods are not trained. As shown in Table 2 of the revised paper, our DPSNet shows promising results on these measures, compared to the state-of-the-art methods. In particular, DPSNet outperforms MVSNet in all metrics, except for the photometric error of filtered MVSNet. Since MVSNet is not designed for dense depth reconstruction, this evaluation protocol is highly disadvantageous to MVSNet.
> In the next response, we will discuss error measures in the ETH3D benchmark, which are also used for MVSNet evaluation.

---

> ### Author Response · Authors · 2018-11-21
> **Response to Reviewer2 (c)**
>
> 3) Since the method largely improved over their baseline algorithms, and the number between different papers are hard to compare. In my opinion, to better show the results, I suggest submitting results to an online benchmark with test data for verifying the results. such as ETH3D multi view benchmark, where everything is standardized.
>
> Different from MVSNet which estimates the full 3D of objects, our DPSNet is inspired by the plane sweeping algorithm which is originally devised for dense depth reconstruction. That is, DPSNet focuses on estimating the dense depth map. By your suggestion, we have submitted depth maps from DPSNet to the ETH3D benchmark, but did not receive satisfactory results, with lower ranks on the ‘F1 score’ metric.
> Method
> (20cm)	low-res many-view	indoor	outdoor	lakeside	sand box	storage room	storage room 2	tunnel
> DPSNet	59.89	54.16	63.70	71.39	70.87	52.09	56.24	48.85
> MVSNet	63.58	56.25	68.47	66	71.12	48.36	64.13	68.29
>
> In order to obtain a high score on the accuracy metric, the depth corresponding to unobserved pixels should be removed. MVSNet performs a depth map filtering to remove outliers, and its multiple observed 3D points merged well to select a correct depth value from multiple observations for a pixel. However, the outlier rejection and the 3D point merging proces are out of the scope of this paper. Instead, we demonstrate that DPSNet produces dense depth maps well by achieving higher scores on the ‘Completeness’ metric than that of MVSNet on the ETH3D benchmark.
> Method
> (20cm)	low-res many-view	Indoor	outdoor	lakeside	sand box	storage room	storage room 2	tunnel
> DPSNet	58.64	47.21	66.25	72.10	77.08	48.12	46.31	49.58
> MVSNet	47.72	40.64	52.43	49.39	55.25	33.58	47.7 	52.66
>
> In conclusion, DPSNet and MVSNet have different goals, and thus different strengths. In the revised paper, we clarify that our DPSNet aims to infer dense depth maps, to highlight its advantages.

---

### Official Review · AnonReviewer1 · 2018-11-02
**Likely Accept: but requires some comments to be addressed**

**Rating:** 7
**Confidence:** 4

**Review:**

The paper describes a method for learning a deep neural network for multi-view stereo. The overall network includes feature-extraction layers applied to all images, followed by a spatial-transformer network (which is differentiable, but with no learnable parameters) that is applied to warp these features from every matching image to the reference image's co-ordinate frame for a series of candidate depth planes, followed by concatenation of the reference and match image features and 3D convolution layers to form a cost volume. The cost volumes of different pairs are averaged, and additional layers are used to refine this cost volume while relying on the reference image's RGB features, followed by soft-max and an expectation over depth values to output the final depth at each pixel. The entire network is trained end-to-end and experiments show that it outperforms state-of-the-art methods for MVS by a significant margin on a number of datasets.

Overall, I have a positive view of the paper and believe it should be accepted to ICLR. However, I would like the authors to address the following issues:

- While the proposed network is complex, I do believe the description of the architecture could be a little better. It would be good to clarify that i indexes view (and N is the total number of views), and provide a few more definitions for the terms in equation (2): namely, are R and t the extrinsics of the reference camera or the i^th camera, etc. The overall approach is clear (for each plane, the method maps features from the paired camera to the reference camera  assuming all points in the the world lie on that plane), but it would be good to clarify the specifics. It might also be useful to emphasize that the cost-volume generation is per-pair (perhaps change the title of Sec 3.2) and that these volumes are averaged for all pairs.

- It might also be useful to apply the algorithm to the rectified binocular stereo case (where the warping and definition of planes by disparity are much simpler), and show comparisons to the many stereo algorithms on datasets like KITTI. At some level, the proposed algorithm can be thought of taking approaches proved to be successful for rectified binocular stereo and generalizing them (by generic warping + plane sweep) to the multi-view case. Hence, such comparisons could be illuminating. (Note: the method doesn't need to outperform the state-of-the-art there, but the results would be informative).

- I do believe the paper would significantly benefit from more discussion of DeepMVS since it's clearly the closest to this method (also solves MVS by deep networks + plane sweep). DeepMVS also learns the matching cost for cost volume generation, and the major difference seems to be that this method is learned end-to-end. It would be better to have a more detailed discussion of the differences (the current discussion at the end of Sec 2 is a little short on details)---architectures, super-vision at the end of the cost-volume vs end-to-end, etc.

---

> ### Author Response · Authors · 2018-11-21
> **Response to Reviewer1 (a)**
>
> 1) While the proposed network is complex, I do believe the description of the architecture could be a little better. It would be good to clarify that i indexes view (and N is the total number of views), and provide a few more definitions for the terms in equation (2): namely, are R and t the extrinsics of the reference camera or the i^th camera, etc. The overall approach is clear (for each plane, the method maps features from the paired camera to the reference camera assuming all points in the the world lie on that plane), but it would be good to clarify the specifics. It might also be useful to emphasize that the cost-volume generation is per-pair (perhaps change the title of Sec 3.2) and that these volumes are averaged for all pairs.
>
> Thanks for your suggestions to improve our manuscript. Following your comments, we have clarified these parts by adding descriptions in Sec 3.2. In the first paragraph of Sec. 3.2, we explain that our cost volume generation is based on unstructured two-view images and can be extended to multiview matching by averaging other cost volumes to reduce a negative effect of image noise. In particular, we have changed the title of Sec 3.2 from “Cost Volume Generation” to “Cost Volume Generation using Unstructured Two-View Images” to highlight that the cost volume is computed from an image pair. We also added explanations for the notations i, R and t in the third paragraph of Sec. 3.2.

---

> ### Author Response · Authors · 2018-11-21
> **Response to Reviewer1 (b)**
>
> 2) It might also be useful to apply the algorithm to the rectified binocular stereo case (where the warping and definition of planes by disparity are much simpler), and show comparisons to the many stereo algorithms on datasets like KITTI. At some level, the proposed algorithm can be thought of taking approaches proved to be successful for rectified binocular stereo and generalizing them (by generic warping + plane sweep) to the multi-view case. Hence, such comparisons could be illuminating. (Note: the method doesn't need to outperform the state-of-the-art there, but the results would be informative).
>
> We applied our algorithm to rectified stereo matching, particularly on the KITTI2015 test set, following the comment of reviewer1. We have obtained reasonable results using the model trained on SUN3D, RGBD, and Scenes11 as shown in Fig. 7(b) and Table 5. We also report the results with the KITTI finetuning in Fig. 7(c) and Table 5.
> Our DPSNet achieves performance compatible to DispNet, but not beyond the performance of recent CNN-based stereo matching algorithms (GC-Net and PSMNet). Based on the rectified stereo experiments, we also agree with reviewer 1’s comments that our DPSNet can be considered a generalized version of stereo matching. We mention that our DPSNet can be directly applied to both rectified stereo and unrectified multiview stereo, and add the experiment results in Sec. 4.3.

---

> ### Author Response · Authors · 2018-11-21
> **Response to Reviewer1 (c)**
>
> 3) I do believe the paper would significantly benefit from more discussion of DeepMVS since it's clearly the closest to this method (also solves MVS by deep networks + plane sweep). DeepMVS also learns the matching cost for cost volume generation, and the major difference seems to be that this method is learned end-to-end. It would be better to have a more detailed discussion of the differences (the current discussion at the end of Sec 2 is a little short on details)---architectures, super-vision at the end of the cost-volume vs end-to-end, etc.
>
> DeepMVS shows good performance on various datasets, but it is not an end-to-end system. DeepMVS computes input volumes for their network using the traditional warping process, different from our network which employs a differential warping process. In addition, even though they take feature aggregations (intra-, inter-feature aggregation), the final depth map is obtained from a conventional CRF which is sensitive to over-smoothing artifacts in textureless regions (see the artifacts in Fig. 3 and Fig. 4 of our paper). Following your comment, we have further explained the difference between DeepMVS and our work at end of Sec. 2.

---

### Public Comment · (anonymous) · 2018-10-05
**Access to Code**

Thanks to the author(s) for this contribution! Are you planning on releasing the PyTorch code for DPSNet?

Also, a implementation question regarding warping the target and reference images to a virtual plane.  At the end of page 3 you describe the formulation for the transformation with respect to the intrinsics, extrinsics, and the particular depth of the virtual plane. You continue to say you use the spatial transform network (STN) to implement this warping. STN defines it transformations with respect to a 2x2 rotation R and 2x1 translation T. What are the values of R and T w.r.t. the intrinsics, extrinsics, and depth?

Thank you again!

---

> ### Author Response · Authors · 2018-10-06
> **Re: Access to Code**
>
> Hello, thank you for your feedback and your interest in our work. Regarding your comments:
>
>
> 1. We will release our PyTorch implementation for DPSNet upon the acceptance of this paper.
>
> 2. The camera rotation R (3x3 matrix) and transform T (3x1 vector) described in our paper are defined in 3D space. Camera extrinsic [R | T] and the intrinsic parameter K are used to obtain the projected image coordinates (\hat{u}_l) described in equation (2). Using the projected image coordinates (\hat{u}_l), we calculate the pixel-wise 2x1 translation vector {T_stn} defined in the STN (similar to the flow field). Because we warp the image based on the projected coordinates, we set the 2x2 rotation matrix {R_stn} defined in the STN as the identity matrix.

---

### Meta-Review · Area_Chair1 · 2018-12-10

**Confidence:** 4
**Recommendation:** Accept (Poster)

**Metareview:**

A deep neural network pipeline for multiview stereo is presented. After rebuttal and discussion, all reviewers learn toward accepting the paper. Reviewer3 points to good results, but is concerned that the technical aspects are somewhat straightforward, and thus the contribution in this area is limited. The AC concurs with the reviewers.